# Estimating Suspended Sediment Concentrations from River Discharge Data for Reconstructing Gaps of Information of Long-Term Variability Studies

**Bárbara M. Jung [1],\*, Elisa H. Fernandes [1], Osmar O. Möller Jr. [1] and Felipe García-Rodríguez [1,2]** 

[1] Instituto de Oceanografia, Universidade Federal do Rio Grande (FURG), CP 474, Rio Grande-RS CEP 96201-900, Brazil; fernandes.elisa@gmail.com (E.H.F.); osmar.moller@gmail.com (O.O.M.J.); felipegr@fcien.edu.uy (F.G.-R.)

[2] Centro Universitario Regional del Este, Sede CURE-Rocha, Ruta 9 s/n, Rocha 27000, Uruguay

\* Correspondence: ibarbarajung@gmail.com; Tel.: +55-27-992421877

**Abstract:** Suspended sediment rating-curves are low cost and reliable tools used all around the world to estimate river suspended sediment concentrations (SSC) based on either linear or non-linear regression with a second variable, such as the river discharge. The aim of this paper is to undertake an evaluation of four different suspended sediment rating-curves for three turbid large river tributaries flowing into the largest choked coastal lagoon of the world, a very turbid system. Statistical parameters such as Nash–Sutcliffe efficiency coefficient (NSE), percent of bias (PBIAS) and a standardized root-mean-square error (RMSE), referred to as *RSR* (RMSE-observations standard deviation ratio) were used to calibrate and validate the suspended sediment rating-curves. Results indicated that for all tributaries, the non-linear approach yielded the best correlations and proved to be an effective tool to estimate the SSC from river flow data. The tested curves show low bias and high accuracy for monthly resolution. However, for higher temporal resolution, and therefore variability, an ad hoc data acquisition to capture extreme rating-curve values is required to reliably fill gaps of information for both performing modeling approaches and setting monitoring efforts for long-term variability studies.

**Keywords:** rating-curve; regression analysis; river discharge; suspended sediment

## 1. Introduction

The coastal export of fine terrigenous continental material from turbid to marine systems consists of a source-to-sink process, which includes erosion and resuspension mechanisms within watersheds, fluvial transport and discharge throughout river-estuaries, and deposition on the inner continental shelf [1]. Such a process depends on geological, geomorphological, sedimentological, and oceanographic features of the region, but its original magnitude is modulated by rainfall, wind and geological attributes such as catchment grain size composition [1,2].

The riverine suspended sediment transport capacity is variable and depends on both natural and anthropogenic factors [3–8]. The river morphology and its ability to erode and transport suspended fine sediments exert negative impacts on the water quality due to the interaction between water and sediment, siltation of reservoirs, soil loss due to erosion and detrimental effects on leisure activities [9,10]. Therefore, estimating the sediment load in turbid systems is a critical aspect at the catchment level [11].

Identifying patterns of the riverine dynamics and variability is very difficult, especially in coastal rainy areas because of increased erosion [12]. One of the main shortcomings in studying the riverine capacity/load of suspended sediment transport is the lack of long-term series of suspended sediment concentrations (SSC) due to the difficulties in installing and maintaining continuous monitoring

stations [13]. For most rivers, however, there is a well-known relationship between water discharge, SSC, water level, depth, and stream velocity [10,11,14,15]. However, the data-sampling interval is a key-issue for running time series analyses to identify oscillation cycles in SSC transport and for development of reliable predictive models.

The limitations of the application of the rating-curve method for three different rivers were analyzed by [9]. The author inferred through a linear least square regression on log-transformed data that datasets with different temporal scale showed significant differences. For seasonal data, the estimated SSC overestimated observed data by 112%. For monthly data, the errors could account for up to +900%. According to the author, this error can be reduced up to +26% by adding evaluation factors such as the flow rate to the regression.

Therefore, in order to develop reliable predictive models, the rating-curve method should be related to a normal discharge regime and must be able to capture high and low variability as well. According to [16], under extremely high discharge conditions, e.g., flood events, changes in the rating-curve parameters become distinctly evident. A different approach to overcome this issue is the use of a flow-weighted or a seasonal-weighted analysis for rating the SSC, which might provide better results for such time-dependent changes [17,18]. A rating curve correction factor was applied by [19] to reduce errors in the log-transformed linear regression, which is a widely used approach for estimating the SSC. The author reported that the error increased when data were back transformed from logarithmic basis, leading to an underestimation of 50% in the SSC. Hence, the application of a factor related to the mean square error of the logarithmic regression was suggested to reduce such underestimation. The use of suspended sediment rating-curves to either monitor or fill in gaps of information in SSC time series is particularly useful under conditions of economical restrictions to set gauging stations [14].

Particularly in humid regions such as the east coast of South America, the continental input material into the ocean attains very large amounts of suspended sediments. In Brazil, coastal lagoons range from small systems, such as Tramandaí-Imbé Lagoon, Peixe Lagoon, Conceição Lagoon, to large systems such as Patos Lagoon (PL). Choked coastal lagoons exhibit high water residence time leading to the accumulation of nutrients, sediments, and organic matter, which are exported to the adjacent ocean [20] The Patos Lagoon (Figure 1), located in southern Brazil, is extremely turbid because of the high runoff input of fine suspended sediments [21,22], which is subject to interannual variability related to El Niño-Southern Oscillation (ENSO) cycles [23–25]. Since there is a large gap of historical SSC data for the system, it is not possible to either perform the calculations to assess the hydro-sedimentological balance or estimate a realistic amount of fine suspended sediment export flowing through the lagoon to the coastal zone [21]. Furthermore, it is very difficult to perform long-term studies of SSC variability and its relationship with ENSO cycles. This information is also most important because once the suspended sediment reaches the estuary, it is partially deposited in the harbor area, demanding periodic dredging operations [7], and the remaining is exported towards the coast through the Patos Lagoon coastal plume [26,27], forming nuisance muddy deposits at the coastal zone and reaching Cassino Beach [21,28].

This study aims to evaluate and identify the best methodological approach to calculate SSC for the Patos Lagoon at the catchment level using the suspended sediment rating-curve method. The behavior of four rating-curve methods, i.e., two non-linear, first and second power function, and two linear, with and without a correction factor (CF), were assessed to select the most accurate approach for forthcoming SSC surveys and siting monitoring stations. River discharge and associated suspended sediment concentrations from the three large sub-catchment tributaries are fundamental as input for modeling fine suspended sediment export processes from Patos Lagoon to the inner shelf and for evaluating the impact of climate change in this coastal area.

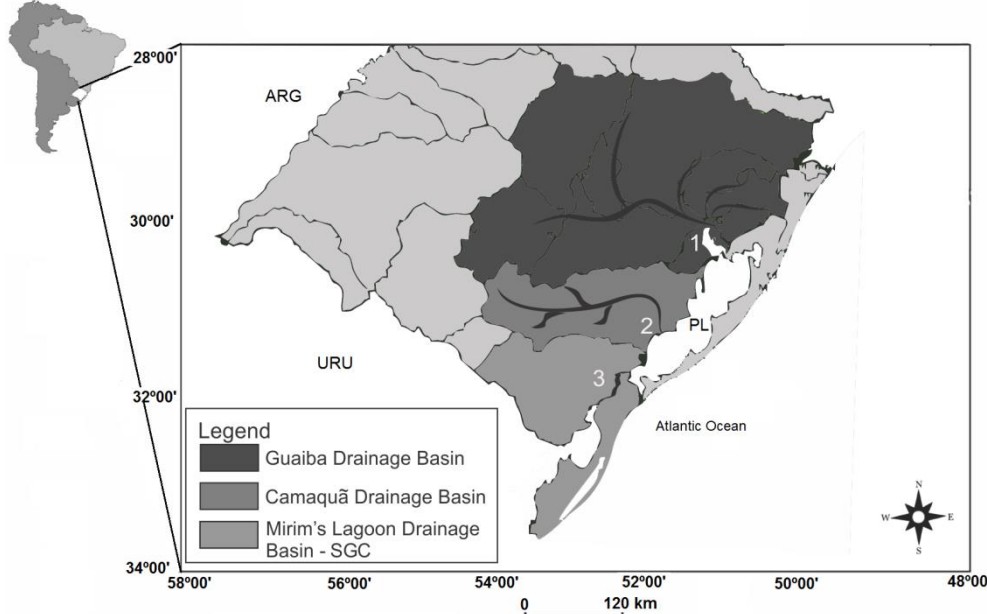

**Figure 1.** Main three sub-catchment tributaries flowing into Patos Lagoon. Different grey scales depict the different drainage basins, identifying the tributaries by numbers. 1. Guaíba Complex; 2. Camaquã River; 3. São Goncalo Channel (SGC).

## 2. Study Area

The Patos Lagoon (Figure 1) is the largest choked coastal lagoon in the world [20], with a narrow channel about 20 km long and 1 km wide compared to the average width of its water body (40 km). The lagoon main tributaries present seasonal variations in their discharge [29,30], with higher values occurring from late August to October (late winter and early spring in Southern Hemisphere) and carrying high concentrations of suspended sediment. Its main tributaries are the Guaíba River at the north, which receives waters from a complex formed by five rivers (including Taquari and Jacuí Rivers, which are responsible for 85% of all the water volume introduced into Guaíba River [31], Camaquã River (central lagoon), and São Gonçalo Channel (SGC), which connects the Mirim Lagoon to the Patos Lagoon (Figure 1). According to [32], the total monthly discharge of the three main rivers to the Patos Lagoon is approximately 2400 $m^3 \cdot s^{-1}$, from which the Guaíba complex corresponds to half of it (average flow of 1253 $m^3 \cdot s^{-1}$).

The Guaíba catchment is the largest of the county, which, together with the Camaquã River drainage, corresponds to an approximate area of 132,000 $km^2$ (Figure 1). This area is highly populated and concentrates intense industrial, agricultural, and commercial activity, which has direct influence in the amount of suspended sediments carried by the rivers. Part of this basin—including the areas along the margins of Patos and Mirim lagoons—is responsible for more than 40% of the total rice derived from inundated crops produced in this county. The period of cultivating rice coincides with the rainy/flood season, making erosion of river margins easier.

The morphological and the sedimentological features of Patos Lagoon have been described by [1,20]. The bottom sediments are distributed as: (1) sandy sediments in the lagoon's margins; and (2) muddy sediments (silt and clay) in deeper portions (central regions and channels). Silt (80%) and clay (15%) are the main sediment types observed in suspension in Patos Lagoon, and they come from the watershed and from wind-wave resuspension [1].

The suspended sediments coming from Guaíba River present a residence time of about 108 days, and the sedimentation rates show a decreasing north-to-south pattern, indicating a strong influence of the Guaíba River discharge on the transport of sediments into the lagoon, which presents an interannual signal related to ENSO events [33]. Previous overall sedimentation rates ranged between 1 and 10 $mm \cdot yr^{-1}$ [34–36], but thereafter [33] reported further consistent sedimentation values

of 7 mm·yr$^{-1}$ in the north, 5 mm·yr$^{-1}$ in central, and 4.8 mm·yr$^{-1}$ in the southern region of Patos Lagoon. The increasing net deposition rates in the last 150 years with values of 3.5 to 8.0 mm·yr$^{-1}$ could be attributed to deforestation and increased erosion in the drainage basin since the European colonization [1]. However, these rates are correlated to the freshwater residence time, which is the time required to renew the total volume of freshwater within a water body [37]. For Patos Lagoon, this time was calculated by [38] using an annual mean discharge of 1000 m$^3$·s$^{-1}$, resulting in a residence time of 1.5 years. According to the author, the higher the river discharge is, the smaller the Patos Lagoon residence time is. Furthermore, [39] calculated the residence time using a two-dimensional depth-averaged finite element flow model with freshwater discharges of 5000, 8000, and 10,000 m$^3$·s$^{-1}$, which resulted in residence times of 135, 85, and 68 days, respectively. Thus, the residence time is controlled mainly by the river flow and its variability directly influences the suspended sediment transport, with accurate measurements being necessary for empiric calculations.

## 3. Methods

In order to determine the best rating-curve method for each of the Patos Lagoon tributaries (Figure 1), time series of SSC and river discharge for each of them were performed. For Guaíba and Camaquã Rivers, weekly time series of SSC and freshwater discharge from 1989 to 1990 were used [40]. For SGC, daily data obtained from 2009 to 2014 were used [29,41]. Although SGC is not characterized as a hydraulic channel, its discharge is governed by wind and water level difference between Mirim Lagoon and Patos Lagoon [42]. According to [43], in the last 100 years, there was a water level increase between both lagoons, which could enhance the flow between them. Hence, the SGC discharge data used for the rating-curves approach were obtained through a slope-type analytical model proposed by [41] using wind and water level time series. Different time frames were used for the calibration and the validation of the four rating-curves with observed data. This approach enables the evaluation of the robustness of each rating-curve under different discharge events [18]. Other sources of data were consulted, such as those from the National Water Agency (ANA) and the Municipal Department of Water and Sewage (DMAE) from the city of Porto Alegre, but these datasets were unreliable, as they presented too many gaps and low correlation, and introduced too many errors into the results.

Different rating-curve methods are available in the literature, but the most frequently used is a power function rating-curve [9,18,44–48] defined as follows:

$$C = a Q^b \tag{1}$$

where C is suspended sediment concentration (mg·L$^{-1}$), Q is the river discharge (m$^3$·s$^{-1}$), and a and b are coefficients based on a least squares regression. These coefficients can be related to the characteristics of the river basin such as river topography contour lines, runoff, river erosion, and transport power [46].

According to [49], the parameters a and b are not constant and can vary with the non-linearity of the suspended sediment rating-curve. For example, they can be altered by changes in river sediment supply, river discharge rates, or a combination of both; therefore, it is important to consider these fluctuations when analyzing the temporal variation of suspended sediment loads [49]. Studying the main stream of the Yangtze River of China, Yang et al. found that river morphology is also directly associated with parameters a and b of the rating-curve [16].

Another approach for estimating suspended sediment concentrations is to add a constant (p) to the above method derived from a second power function [18]. The author tested several different rating-curves with different data assemblies using a power function with an additional constant term p. The method is based on non-linear least square regression of both discharge and suspended sediment, presenting the best fit for the data:

$$C = a Q^b + p \tag{2}$$

When working with a least squares fitting curve, one of the main disadvantages is the sensitivity to outliers. To overcome this, a least absolute residual (LAR) technique was used. This method adds robustness to the regression and helps to alleviate the retransformation bias [50]. Although most researchers use these power function equations (Equations (1) and (2)), other methods can be suggested to estimate SSC [18,49,51,52]. A different but widely used method is the logarithmic regression (Equation (3)), which works with linear regression, a simpler and objective approach [16,49,52]. The SSC and the discharge time series are log-transformed in order to obtain the linearity, which may be considered more dynamic [9,44,45].

$$\log(C) = \log(a) + b \, \log(Q) \tag{3}$$

The accuracy of a linear and a non-linear approach to estimate suspended sediment loads was compared by [53] using the Big Blue River and the Wabash River data. The author argued that a linear regression represents more accurately the estimates of SSC due to the residual error distribution behavior of the non-linear approach. Despite the advantage of being simpler, log-based regressions tend to underestimate the SSC values. Thus, a correction factor (CF) is used for the bias introduced by the log-transformation when the values are back-transformed [19,54]:

$$CF = \exp\left(2.65 \, \sigma^2\right) \tag{4}$$

The CF is related to a mean squared error derived from the log-transformed regression ($\sigma^2$). Several authors [18,51,52,55] discussed the use of different rating-curve methods to determine the best fit for each river. Rating-curves were tested considering the complete dataset and using monthly averages. Statistical models were used to verify the accuracy of each method in performing realistic predictions and the best rating-curve for each river.

In order to build a regression curve for estimating SSC from discharge data, we first performed correlation analyses. For all datasets, the Pearson's correlation coefficient between the discharge and the SSC data was determined at 0.05 level of significance. Besides Pearson's correlation coefficient (R), basic to any predictive model that describes the direct correlation between simulated and observed data, other parameters recommend by [56] were used for both the calibration and the validation of the rating-curves. These are the RSR (RMSE-observations standard deviation ratio), the Nash–Sutcliffe efficiency coefficient (NSE) [57], and the percent of bias (PBIAS) [58]. Another suggested factor to compare the accuracy of the ratting curves in estimating SSC loads is the root-mean-square error (RMSE) [59]. This method has, however, limitations such as its sensitivity to outliers, which can induce to some errors when comparing different curves. The RSR method recommended by [56] works with a scaling/normalization factor, which uses the ratio of RMSE and the standard deviation of the observed data to establish a level of comparison between the rating-curves.

The NSE is related to a goodness-of-fit index and evaluates the magnitude of the residual variance (noise) of the predicted value compared to the measured data variance [57]. This coefficient indicates the accuracy of the simulated data in representing the observed data (Equation (3)), which can be described as:

$$NSE = 1 - \left[ \frac{\sum_{i=1}^{n}\left(C_i^{obs} - C_i^{sim}\right)2}{\sum_{i=1}^{n}\left(C_i^{obs} - C_i^{mean}\right)2} \right] \tag{5}$$

where $C_i^{obs}$ is relative to the SSC observed data relative to the same index i of SSC simulated data ($C_i^{sim}$). The $C_i^{mean}$ is the mean SSC observed data from n observations. Values can range from $-\infty$ to $+1$, where values close to $+1$ are considered as optimal, those between 0.0 and $+1.0$ are acceptable, and those values smaller than 0.0 mean that the noise from the model overlaps the real data. In other words, for values close to zero, the model is inefficient in reproducing the reality, and the use of an average value is more efficient to represent the local SSC value [56]. As an evaluation criterion, the NSE has an ideal value ranging close to $+1$.

After determining the correlation between the freshwater discharge and the SSC time series, the next step was to investigate the response of each rating-curve method in predicting the SSC for the main river tributaries. The PBIAS gives an idea about the range of the prediction. It measures the tendency of the simulated data being higher or lower than the observed data, giving an indication of the model performance:

$$PBIAS = \frac{\sum_{i=1}^{n}\left(Ci^{obs} - Ci^{sim}\right) * 100}{\sum_{i=1}^{n}\left(Ci^{obs}\right)} \tag{6}$$

An optimal value for PBIAS is 0.0. Positive values indicate underestimation, and negative values indicate overestimation of the simulated data compared to the observed data [58]. For the rating-curves calibration and validation, the evaluation criteria were used as indicated by [56], which classifies the curves performance as "Very Good", "Good", "Satisfactory", and "Unsatisfactory". The ranges of each criterion are presented in Table 1.

**Table 1.** Calibration and validation criteria, adapted from [56].

|  | RSR | NSE | PBIAS |
|---|---|---|---|
| **Very Good** | 0 to 0.5 | 0.75 to 1.0 | < ± 15 |
| **Good** | 0.5 to 0.6 | 0.65 to 0.75 | ± 10 to ± 30 |
| **Satisfactory** | 0.6 to 0.7 | 0.5 to 0.65 | ± 30 to ± 55 |
| **Unsatisfactory** | >0.7 | <0.5 | > ± 55 |

RSR: RMSE-observations standard deviation ratio; NSE: Nash–Sutcliffe efficiency coefficient; PBIAS: percent of bias.

The curves were performed and calibrated using an annual dataset for each tributary and validated with the subsequent year dataset. The statistical parameters were used for both calibration and validation in order to achieve the highest possible accuracy. Curves 1 and 2 are based on non-linear regression, with and without a constant term (p), respectively. Curves 3 and 4 are linear log-transformed regressions with and without the CF factor, respectively.

## 4. Results

Different approaches used to estimate SSC for Patos Lagoon tributaries yielded satisfactory correlation coefficients. For all datasets, the correlation between discharge and SSC data was higher than 70%, except for Camaquã River that yielded values lower 70%. Logarithmic transformation of SSC data was evaluated as a valid option to go over samples for asymmetrical data distributions in order to explore data correlation behavior. Differences between correlations for log-transformed (Log_Based) data ranging from 68% to 95% confirmed the strong relationship between variables (Table 2).

**Table 2.** Pearson correlation for suspended sediment vs. river flow.

|  | Normal | Log Based |
|---|---|---|
| Guaíba—All Data | 0.76 | 0.77 |
| Guaíba—Monthly Average | 0.96 | 0.95 |
| Camaquã—All Data | 0.47 | 0.68 |
| Camaquã—Monthly Average | 0.87 | 0.84 |
| SGC—All Data | 0.80 | 0.89 |

For both Guaíba and Camaquã Rivers, the use of monthly averaged data increased the correlation between the SSC and the river discharge time series. This behavior is expected because monthly averages remove high frequency variability from the dataset. These correlations are in agreement with literature for other areas [11,45,60]. Monthly averaged curves were not applied for the SGC on account

of unreasonable results for further evaluation. [11] also used successfully mean daily water discharge and sediment discharge for different periods of available data to study spatial and temporal variations of rating curves in the Chinese Loess Plateau.

Most of the curves displayed a similar concave shape, and their behavior was similar for fitting the data scattering (Figure 2). The Guaíba rating-curves presented the best fit to the observed data, where the four different curves (Figure 2a,b) showed similar behavior for both the whole dataset and the monthly averaged data. However, the Camaquã rating-curves presented a different behavior. For higher discharges, the curves tended to diverge in particular ways (Figure 2c,d). For discharges lower than 700 $m^3 \cdot s^{-1}$, however, the rating-curves could be reliably applied. Most of the SGC curves tended to overlap, with exception of Curve 4, which diverged exponentially along with the discharge changes as a result of an unsatisfactory calibration of the method (Figure 2e). As observed by [18], the power function regression seems to visually better fit the data in all cases. The a, b, p, and CF parameters used to build each rating-curve are presented in Table 3.

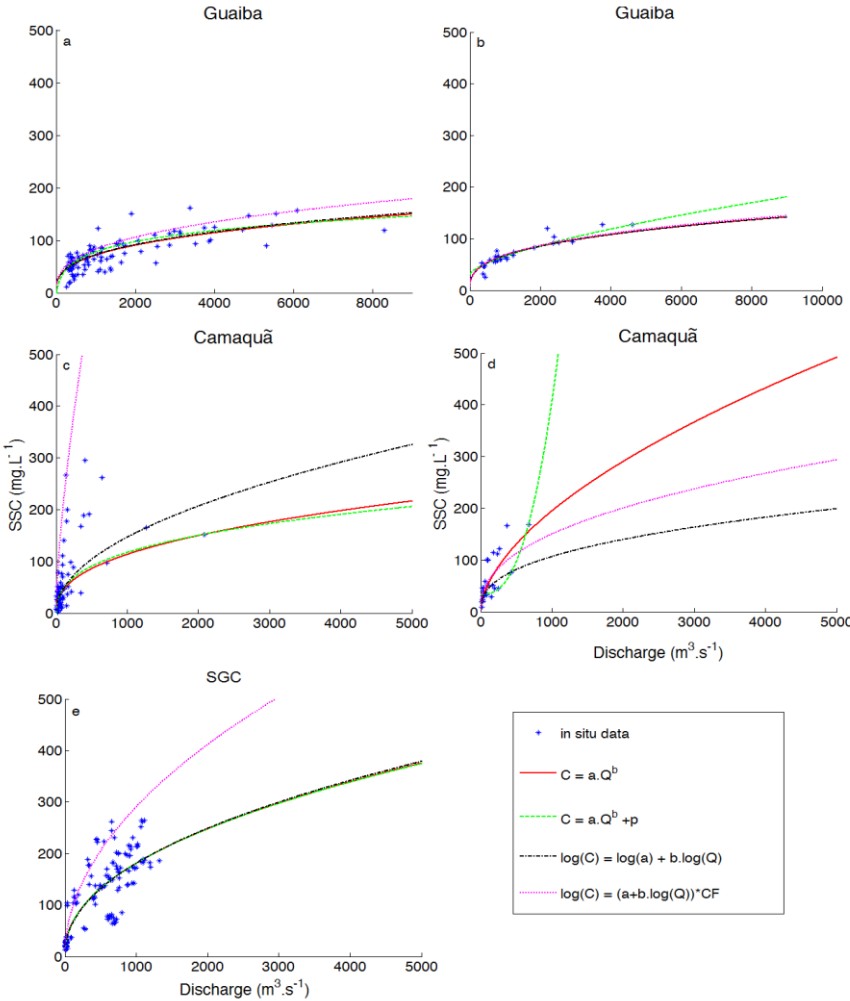

**Figure 2.** All data (left) and monthly average (right) rating-curves: (**a**) Guaíba all data rating-curve; (**b**) Guaíba monthly data rating-curve; (**c**) Camaquã all data rating-curve; (**d**) Camaquã monthly data rating-curve; (**e**) SGC all data rating-curve.

**Table 3.** Suspended sediment rating-curves coefficients and statistical parameters. Best results are presented in bold. CF: correction factor.

| | Coefficients | | | | Calibration | | | Validation | | |
|---|---|---|---|---|---|---|---|---|---|---|
| | **a** | **b** | **p** | **CF** | **RSR** | **NSE** | **PBIAS** | **RSR** | **NSE** | **PBIAS** |
| **Guaíba—All data** | | | | | | | | | | |
| **Curve 1** | **7.09** | **0.34** | **-** | **-** | **0.60** | **0.64** | **−1.49** | **0.60** | **0.64** | **−2.56** |
| Curve 2 | 81.95 | 0.13 | −125 | - | 0.63 | 0.61 | −6.92 | 0.62 | 0.62 | −8.28 |
| Curve 3 | 0.84 | 0.34 | - | - | 0.60 | 0.64 | −2.20 | 0.60 | 0.65 | −3.28 |
| Curve 4 | 0.84 | 0.34 | - | 1.03 | 0.77 | 0.40 | −17.49 | - | - | - |
| **Guaíba—Monthly Average** | | | | | | | | | | |
| Curve 1 | 6.48 | 0.34 | - | - | 0.31 | 0.90 | 2.39 | 0.45 | 0.80 | 0.33 |
| Curve 2 | 0.36 | 0.66 | 30.17 | - | 0.29 | 0.92 | 1.72 | 0.41 | 0.83 | −0.50 |
| Curve 3 | 0.81 | 0.34 | - | - | 0.31 | 0.90 | 2.40 | 0.45 | 0.8- | 0.33 |
| Curve 4 | 0.81 | 0.34 | - | 1.00 | 0.30 | 0.91 | 0.69 | 0.44 | 0.80 | −1.40 |
| **Camaquã—All data** | | | | | | | | | | |
| Curve 1 | 7.00 | 0.40 | - | - | 0.85 | 0.27 | 25.14 | - | - | - |
| Curve 2 | 16.54 | 0.31 | 21.51 | - | 0.84 | 0.30 | 25.03 | - | - | - |
| Curve 3 | 0.68 | 0.50 | - | - | 0.83 | 0.31 | 19.24 | - | - | - |
| Curve 4 | 0.68 | 0.5 | - | 1.38 | 4.83 | 22.36 | 299.7 | - | - | - |
| **Camaquã—Monthly Average** | | | | | | | | | | |
| **Curve 1** | **3.68** | **0.58** | **-** | **-** | **0.54** | **0.71** | **−4.08** | **0.68** | **0.54** | **16.09** |
| Curve 2 | 8.3E-6 | 2.55 | 33.30 | - | 0.50 | 0.75 | 10.60 | 1.06 | −0.13 | 41.38 |
| Curve 3 | 0.86 | 0.39 | - | - | 0.70 | 0.52 | 18.73 | 0.89 | 0.20 | 32.90 |
| Curve 4 | 0.86 | 0.39 | - | 1.07 | 0.57 | 0.68 | −8.09 | 0.70 | 0.51 | 11.00 |
| **São Gonçalo Channel—All data** | | | | | | | | | | |
| **Curve 1** | **7.88** | **0.45** | **-** | **-** | **0.58** | **0.66** | **0.87** | **0.60** | **0.65** | **4.38** |
| Curve 2 | 8.12 | 0.45 | 1.04 | - | 0.58 | 0.66 | 0.98 | 0.60 | 0.65 | 4.49 |
| Curve 3 | 0.88 | 0.46 | - | - | 0.58 | 0.66 | 1.28 | 0.60 | 0.64 | 4.81 |
| Curve 4 | 0.88 | 0.46 | - | 1.09 | 1.18 | −0.39 | −54.83 | - | - | - |

In order to quantify the fit of each rating-curve to the measured data, statistical parameters were calculated during the calibration procedure (Table 3), and the best curves are highlighted in bold, while the unsatisfactory values are underlined. Results indicate that monthly averages and Curve 1 produced a better fit for Camaquã suspended sediment rating-curve, and the use of all data and Curve 1 performed better for Guaíba and São Gonçalo Channel.

Along with the PBIAS, the RSR represents the error input from the rating-curve method to the calculated data. The lower the value of these parameters is, the more efficient the rating-curve approach is. The curves that did not present at least a "Satisfactory" classification mark for calibration were not validated (Table 3).

It is noticeable from the results (Table 3) that these three parameters behaved differently for each dataset, and the best fit for one parameter did not necessarily imply the best fit for the other two. For Guaíba River, the measured data presented an SSC average of 73 mg·L$^{-1}$, with an average discharge of 1489 m$^3$·s$^{-1}$, and the statistical parameters varied from "Very Good" for monthly averages to "Unsatisfactory" for the "All Data" set using Curve 4. For the "All Data" set, Curve 1 presented the best fit, and Curve 4 exhibited an "Unsatisfactory" performance. For monthly averaged data, all curves presented a "Very Good" performance, with Curve 2 attaining the greater accuracy. Thus, for Guaíba River data, both power function and log-transformed curves presented high efficiency and

low PBIAS. [18,61] commented that the power function used to achieve a higher efficiency than other regression curves is usually the regression curve.

Camaquã River showed a different behavior for each of the rating-curves. The calibration of "All Data" curves failed to reproduce accurately suspended sediment data, resulting in an overall "Unsatisfactory" behavior of the rating-curves, where the higher error was detected for Curve 4, presenting a PBIAS of over 299%. According to [62], the "Unsatisfactory" calibration could be the result of five factors: (a) different datasets for curve calibration and validation; (b) inadequate model calibration; (c) insufficient number of observations; (d) quality; and (e) model unable to reproduce the environmental behavior. Monthly data, on the other hand, presented a "Very Good" to "Satisfactory" classification during calibration. The validation, however, exhibited outliers. With an observed SSC average of 62 mg·L$^{-1}$ and a mean discharge of 153 m$^3$·s$^{-1}$, the power functions with the additional constant and the log transformed curve (Curves 2 and 3, respectively) presented higher errors when validating simulated against observed data. Therefore, the power function was selected as the most accurate suspended sediment rating-curve.

For Guaiba's analysis, the behaviors of the rating-curves were similar to Guaíba River "All Data" set, where Curve 4 was also withdrawn due to its high PBIAS and weak calibration. Among the other curves, the most accurate rating-curve was the power function regression (Curve 1). With a measured SSC average of 118 mg·L$^{-1}$ and water discharge of 496 m$^3$.s$^{-1}$, the SGC presented "Very Good" PBIAS, which indicates low differences between simulated suspended sediment discharge values and measurements.

Thus, results indicated that the power function curve was the most accurate and reliable rating-curve for all three rivers. Despite each rating-curve behaving differentially, they all presented high accuracy and low error. The Guaíba River curve overestimated the sediment load by 0.49% to 2.5%, and the Camaquã River curve underestimated it by approximately 16% and the SGC by 4.37%, which are considered as high-performance outcomes [56]. The observed and the calculated SSC behavior can be observed in Figure 3. The calculated SSC is based on the corresponding best rating curve for each environment. The Guaíba and Camaquã River SSC estimates are based on a monthly approach, while the SSC estimates for the SGC was calculated using all SSC available data.

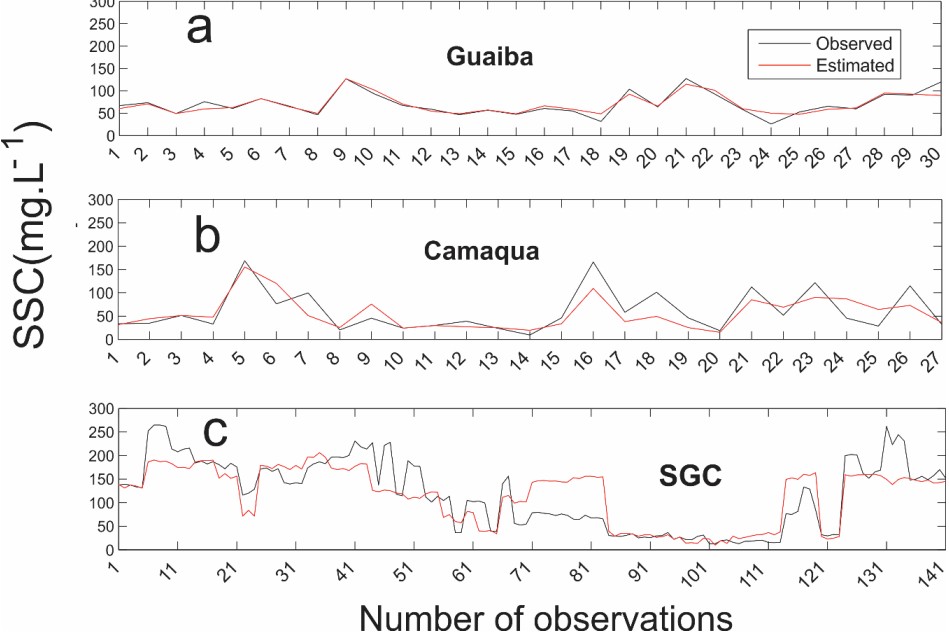

**Figure 3.** Comparison of interannual distribution of observed vs. calculated suspended sediment concentrations (SSC) for (**a**) Guaíba (**b**) Camaquã and (**c**) SGC using the best rating curve approach.

## 5. Discussion

The overall assessment of the suspended sediment rating curve approach for load estimation on Patos Lagoon tributaries presented higher performance than other rivers, such as those presented by [10]. The rating-curves used by them presented maximum errors of −76% to +63% on the River Bandon with errors of −65% to +359% found on the River Owenabue. The authors observed that the most accurate load estimate for the Bandon River was obtained using a stage separated power curve, while for the Owenabue River, the most accurate load estimate was obtained using a general power curve. The analysis of Patos Lagoon tributaries, for all datasets indicated that the power function (Curves 1 and 2) presented the best results, with low PBIAS error and a high curve calibration. For Guaíba River, the annual suspended sediment discharge, calculated from the SSC best rating-curve and associated mean river discharge, was approximately of $2.99 \times 10^6$ ton·yr$^{-1}$, with a 0.5 to 2.5% overestimation bias. For Camaquã River, the suspended sediment discharge was about $0.24 \times 10^6$ ton·yr$^{-1}$, with an underestimation of 16%. For the SGC, the discharge was around $1.88 \times 10^6$ ton·yr$^{-1}$ and presented an underestimation of 4.4%. Although Camaquã River presented the higher PBIAS among the others (16%), its bias is still considered small in comparison with the 55% bias for a "Satisfactory" classification of suspended sediment [56].

The statistical parameters analyzed (i.e., RSR, NSE, and PBIAS between simulated and observed data) were satisfactory and enabled us to select the best approach for each of the Patos Lagoon main tributaries. Several authors have also used at least one of these parameters to evaluate regression curves and obtained similar results [18,50–52,56]. For example, Heng and Suetsugi tested the power function rating-curve at 16 different sub-catchments along Lower Mekong Basin, and their results presented a mean PBIAS of 6.44%, RSR of 0.61, and NSE of 0.63, which are similar to Guaíba, Camaquã, and SGC results [13]. Regarding both Guaíba and Camaquã Rivers, the use of monthly averaged data increased the correlation between SSC and river discharge time series. This behavior is expected since monthly averages remove high frequency variability from the dataset. These correlations are in agreement with literature for other areas [11,45,60]. A high correlation coefficient between datasets allow one to consolidate the performance of rating curves. According to [18], when organizing the data into longer time scales of variability, such as seasonal, high and low discharge, and annual loads, the efficiency of the rating-curves increases, and the SSC prediction becomes more accurate. Therefore, for the dataset evaluated in this paper, the Guaíba and Camaquã rating curves improved their performance with decreasing temporal resolution. Monthly data presented smaller errors and a calibration and higher correlation coefficients. This fact can be clearly observed in the Guaíba rating-curves, in which monthly averages PBIAS tended to be lower than those of the entire dataset.

Similar results were also reported by [51] when testing different time resolution for suspended sediment rating-curves, as the suspended sediment curve errors can increase for short-term-frames, such as daily and weekly data, due to the high frequency/variability of the events, which are filtered when long-term processes are assessed (e.g., monthly or annually. In addition, it's also reported that the difference in accuracy of daily and even higher resolution sampling efforts for a monthly/annual sediment rating-curve do not justify the high cost and the time-consuming work involved in a daily record gauging station [63]. It cannot be overlooked that differences in rating curves tying river discharge to any other variable are also dependent on the quality of the original rating curve that provides discharges from measured water level [64], in most cases in Brazil, twice a day by a single observer.

Changes in suspended sediment rating-curves and the measured riverine SSC are not only caused by the river discharge itself but also by other environmental aspects [49,65]. However, sediment rating-curves allow one to reliably calculate the sediment supply budget from river discharge (especially under economical restrictions) to better understand the environmental dynamics in view of maintenance and/or rehabilitation.

Patos Lagoon carries high concentrations of suspended sediments into the southern Brazilian inner continental shelf, causing several problems such as mud-flows and muddy deposits along the coast,

directly affecting regional economy [21]. Despite the importance of this region, there are no available studies on the Patos Lagoon hydro-sedimentological export to the coastal zone. Thus, the main goal of this paper was to elaborate an initial reliable tool to overcome the lack of suspended sediment data.

Once the best suspended sediment rating-curve was reliably established for each of the main Patos Lagoon tributaries, it was possible to reconstruct long-term data of SSC by filling in the gaps for which only water discharge data were available. This type of approach is used around the world due to the high cost of maintaining gauging stations and the handwork of recording SSC. The time step and the number of samples needed to calculate an efficient rating-curve depend on the desired monitoring temporal resolution.

According to [51], for long time scales such as annual exporting rates, a few SSC samples would be enough for a reliable estimate. For higher temporal resolution of total SSC, such as monthly or weekly exporting rates, however, more samples would be required to validate the rating curve. In our case, for an efficient annual estimation, no more than 12 samples per year are necessary, or for a 5-year estimative, six samples per year appear to be acceptable to produce an accurate rating-curve. Only the Guaíba data enabled the comparison between different time interval curves. Camaquã River and the SGC yielded non-acceptable results for comparison, because, for Camaquã River, the "All Data" curve was ranked as "Unsatisfactory", while for SGC the monthly curve presented unsuitable results to perform an accurate curve analysis. The Guaíba River calibration and validation curve made it possible to conclude that, for annual loads, the monthly curve presented the smaller errors and the higher correlation between simulated and observed data. This result is consistent with other authors who observed that daily records are not necessary to perform monthly to annual rating-curve estimates [51,63].

Rating-curves can be used not only for the main tributaries but for secondary streams as well. By creating an interconnected network of Patos Lagoon basin, it would be possible to optimize the calibration and the validation of hydro-sedimentological models for the entire system and achieve a more comprehensive understanding of Patos Lagoon sedimentary balance by comparing modern sedimentation rates [66] and the sediment output to the ocean [27]. In this way, one could calibrate a large-scale regional model to estimate an integrated sedimentological balance of Patos Lagoon and determine the export rates to the adjacent ocean.

## 6. Conclusions

The main tributaries contribute approximately $5.1 \times 10^6$ ton·yr$^{-1}$ of suspended sediments into Patos Lagoon. Part of the fine fraction deposits within the lagoon or the estuarine region, and that remaining is exported to the inner shelf. The hydrological non-linear approach yielded reliable efficiency and can be used to calculate either monthly or annual SSC. Further data are necessary to continue recalibrating the suspended sediment rating-curve and improving temporal resolution. For hindcasting calculations, this approach may not change the accuracy of the suspended sediment rating-curve, but for forthcoming years, it is important to look for changes in the curve efficiency, as population development and growth can dramatically change land use and soil loss due to paddy fields in the region.

Although river flow records are available to reconstruct long-term SSC variability, it is advisable to continue measuring direct suspended sediment data at least on a monthly basis to keep improving the calibration of curve performance. Once curves no longer show significant long-term changes, they could be used to investigate decadal scales of variability related to the climate variability that intensify rainfall and stream flow along the seasons and the influence of El Niño and La Niña cycles.

**Author Contributions:** B.M.J. contributed with conceptualization, methodology, formal analysis, writing original draft; E.H.F. contributed with conceptualization, review and editing the text, and supervision; O.O.M.J. contributed with conceptualization, data curation, review and editing the text, and supervision; and F.G.-R. contributed with conceptualization and review and editing the text. All authors have read and agreed to the published version of the manuscript.

**Funding:** This research was funded by FINEP through the TRANSAQUA Project (grant 01.11.0141.01) and REHMANSA Project (grant 01.12.0064.00); by CNPq through research grants 551436/2011-5 (EHF), 307602/2014-1 (EHF), 302586/2019-9 (OMJ) and 304007/2019-6 (FGR), and by the Office of Naval Research, USA, which sponsors the LOAD Project (contract N62909-19-1-2145).

**Acknowledgments:** The authors would like to acknowledge CAPES for sponsoring the first author's (BMJ) Master's Degree. Thanks also to DMAE–Departamento Municipal de Água e Esgoto from Porto Alegre city and to ANA–Agência Nacional de Águas for data river discharge data.

**Conflicts of Interest:** The authors declare no conflict of interest.

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
