# Peer review of "Estimating Suspended Sediment Concentrations from River Discharge Data for Reconstructing Gaps of Information of Long-Term Variability Studies"

_water, doi:10.3390/w12092382_

Round 1
Reviewer 1 Report
This is very interesting work and I love reading this paper. Each section is well explained.
I recommend for its publication with minor revision after responding to following points:
1. Please provide reference for the sentence "The coastal exportation...." in line number 29-32.
2. For table number 2, can you please explain how did you calculate log based values in this table?

Reviewer 2 Report
General comments:
This paper presents discharge (Q) and suspended sediment concentration (SSC) regression analysis for three rivers in the Patos Lagoon in Brazil. Several regression relationships are applied to original and log-transformed data, and to all data and monthly averaged data.
It is interesting to see the results of the work presented. However, it is not clear what new scientific contribution this work presents. As is clear from the literature (that the authors properly cite), a lot of work already exists on the relationship between Q and SSC. What is the transferable value of this work?
In the results section, authors present data averages (both for discharge and concentration) and sediment load far beyond their digital significance. Please present data with the appropriate number of significant digits.
In Figure 2, why are the x-axis ranges for the Camaqua and SGC so far beyond the maximum observed discharge values? Also, it looks as if a linear fit to the Camaqua data set is plausible, or that perhaps some hysteresis effect is present.
The discussion section can be shorted, as a lot of the text is somewhat repetitive about the goodness of fit of the different regression types on different data types. While some comparison with the literature is necessary, authors should try to be more concise and add what is really new for publication. The idea that fewer samples per year are sufficient to estimate the sediment load goes somewhat against other literature that recommends capturing high load events.
The manuscript is well organized and mostly well written. However, I suggest further attention to English writing that was not addressed in this review.
Detailed comments:
I have provided editorial assistance to improve the English writing in the Introduction. I suggest that the authors invest a little more time to improve the English writing in the remainder of the manuscript.
Line 16: evaluate use of 'choked' (not 'chocked' which must be a spelling error)
Line 19: change 'indicated' to 'indicate'
Line 20: change 'proved' to 'prove'
Line 21: change 'showed' to 'show'
Line 30: remove 'the' (words in plural often do not need the article; please check throughout)
Line 39: suggest to replace 'fundamental' with 'critical' or 'key'
Line 40: insert 'the' before 'catchment'
Line 44: replace 'setting' with either 'siting' or 'installing' depending on what is the perceived challenge. Siting would main it is hard to find suitable locations for stations. (see also line 84)
Line 45: replace 'current' with 'stream'
Line 47: replace 'come to' with 'for development of'
Line 50: suggest to change 'there were significant differences when using different time scale data sets' with 'data sets with different temporal scale showed significant differences'
Line 51: change 'accounted for up to 112% overestimation in comparison with in situ data' to 'overestimated observed data by 112%'
Line 54: change 'elaborate' to 'develop' and 'predicting' to 'predictive'
Line 71: change to 'historical SSC data'
Line 73: change 'throughout' to 'flowing through'
Line 74: delete second 'the' (before SSC)
Line 75: change 'their' to 'its'
Lines 67-79: the focus here on South America is okay. However, there are many examples where sediment transport chokes surface water systems, so highlighting only the Patos Lagoon may be somewhat limited.
Line 81: insert 'the' before 'catchment' and change 'through' to 'using'
Line 83: change 'the correction factor' with 'a correction factor'
Line 84: delete 'The', then capitalize 'River'
Line 85: change to 'three large sub-catchment tributaries'
Line 86: delete 'an' and 'the' before the word 'fine'
Line 87: change to 'climate change'
Line 100: is it relevant for the reader to know that the Guaiba drainage basin is the largest in the county? (you could use 'catchment' instead of 'drainage basin', consistent with use in Introduction)
Line 130: insert 'analyses' after 'time series'
Line 144: change 'found' to 'available'
Line 150: add s to make it 'least squares regression' (same in line 162, and other places)
Line 158: change 'the parameter' to 'parameters'
Line 163: change 'for' to 'to'
Line 198: it is unclear what you mean by 'induce some errors when comparing different curves'
Line 211: change 'as optimal values' to 'as optimal'
Line 224: please reword 'it was used the [56] evaluation criteria'
Line 230-233: I suggest moving this to 'Methods'. Start the paragraph with the main result, which is line 233-235. Merging lines 233-235 with paragraph 239-245 would be possible.
Line 234: change 'higher' to 'greater' (note that for Camaqua only the normal data were below 50%; did you mean to write 'below 70%'?)
Line 246-254: almost this entire paragraph is Methods, not result. Only the final sentence provides a result statement. You should move 246-253 to Methods, and merge 253-254 as the topic sentence (slightly reworded) for the next paragraph.
Line 255: 'best fitness to measurements' can be replaced with 'best fit to the observed data'
Line 277: no values are underlined. Is this an oversight? – change 'indicated' to 'indicate' (check for this occurrence elsewhere as well)
Line 281: change 'lowest' to 'lower'
Line 290: change 'higher' to 'greater' (same in Line 292)
Reviewer 3 Report
Congratulations. This is a very well-represented, timely and explained research article. My only comment is - If it's possible, try to include some more discussions to explain your research findings. That will help in understanding the significance and novelty of the present research.
Round 2
Reviewer 2 Report
General comments:
In the previous review, I asked what is the transferable value of this work? The answer is not really going beyond the scientific value I was hoping for, but you have provided an improved reason for the publishable value of the work.
The response to my comment about presenting data beyond their digital significance was probably misunderstood. It is not about what is adequate for presentation, but the precision by which discharge and SSC can be measured. Two digits (or three) is fine for regression coefficients in Table 2 and for goodness-of-fit parameters in Table 3. The averages I am referring to are in the text below Table 3 (lines 398-428). For example, in line 400, instead of 72.94 mg/L, report 73 mg/L. and on the next line instead of 1488.8 m3/s, report 1489 m3/s, etc.
The writing has improved as well. I appreciate the focus on topic sentences in the results section.
Detailed comments:
Line 80: replace 'huge magnitudes of SSC' with 'very large amounts of suspended sediments' (please note however that the same is stated in line 85)
Line 82: remove the comma
Line 193: delete the period, and end the sentence with ', defined as follows:
Line 197: 'Where' should be 'where' (it is still part of the same sentence started on line 192); insert 'and' before 'a and b'
Line 300: remove the comma
Author Response
Replay to Comments and Suggestions for Authors (in bold)
General comments:
In the previous review, I asked what is the transferable value of this work? The answer is not really going beyond the scientific value I was hoping for, but you have provided an improved reason for the publishable value of the work.
Thanks for your comment. We understand that these results will be essential for the local management of Patos Lagoon, as it is now possible to calculate the net balance of SSC in the system for the first time. Furthermore, SSC time series with monthly resolution will also be a significant improvement in the resolution of SSC boundary condition in the numerical models developed for the area.
The response to my comment about presenting data beyond their digital significance was probably misunderstood. It is not about what is adequate for presentation, but the precision by which discharge and SSC can be measured. Two digits (or three) is fine for regression coefficients in Table 2 and for goodness-of-fit parameters in Table 3. The averages I am referring to are in the text below Table 3 (lines 398-428). For example, in line 400, instead of 72.94 mg/L, report 73 mg/L. and on the next line instead of 1488.8 m3/s, report 1489 m3/s, etc.- revised as requested
The writing has improved as well. I appreciate the focus on topic sentences in the results section. – we are glad you are satisfied.
Detailed comments:
Line 80: replace 'huge magnitudes of SSC' with 'very large amounts of suspended sediments' (please note however that the same is stated in line 85) - revised as requested
Line 82: remove the comma - revised as requested
Line 193: delete the period, and end the sentence with ', defined as follows: - revised as requested
Line 197: 'Where' should be 'where' (it is still part of the same sentence started on line 192); insert 'and' before 'a and b' - revised as requested
Line 300: remove the comma - revised as requested
